# Facile Gold-Nanoparticle Boosted Graphene Sensor Fabrication Enhanced Biochemical Signal Detection

**DOI:** 10.3390/nano12081327

**Published:** 2022-04-12

**Authors:** Shuaishuai Meng, Li Wang, Xixi Ji, Jie Yu, Xing Ma, Jiaheng Zhang, Weiwei Zhao, Hongjun Ji, Mingyu Li, Huanhuan Feng

**Affiliations:** 1Sauvage Laboratory for Smart Materials, Flexible Printed Electronic Technology Center, School of Materials Science and Engineering, Harbin Institute of Technology (Shenzhen), Shenzhen 518055, China; m15052689951@163.com (S.M.); 18s154783@stu.hit.cn (L.W.); maxing@hit.edu.cn (X.M.); zhangjiaheng@hit.edu.cn (J.Z.); wzhao@hit.edu.cn (W.Z.); jhj7005@hit.edu.cn (H.J.); myli@hit.edu.cn (M.L.); 2School of Materials Science and Engineering, Harbin Institute of Technology (Shenzhen), Shenzhen 518055, China; jixixi201408@163.com (X.J.); jyu@hit.edu.cn (J.Y.)

**Keywords:** 3D graphene, nano-enzyme, gold nanoparticle, nanodoping, multiple-biochemical signal detection

## Abstract

Graphene has been considered as an excellent biochemical sensors’ substrate material because of its excellent physical and chemical properties. Most of these sensors have employed enzymes, antibodies, antigens, and other biomolecules with corresponding recognition ability as recognition elements, to convert chemical signals into electrical signals. However, oxidoreductase enzymes that grow on graphene surfaces are affected significantly by the environment and are easily inactivated, which hinders the further improvement of detection sensitivity and robusticity. A gold-boosted graphene sensor was fabricated by the in situ electrochemical deposition of inorganic gold nanoparticles on vertical graphene nanosheets. This approach solves the instability of biological enzymes and improves the detection performance of graphene-based sensors. The uric acid sensitivity of the gold-boosted electrode was 6230 µA mM^−1^ cm^−2^, which is 6 times higher than the original graphene electrode. A 7 h GNSs/CC electrode showed an impressive detection performance for ascorbic acid, dopamine, and uric acid, simultaneously. Moreover, it exhibited a reliable detection performance in human serum in terms of uric acid. The possible reason could be that the vertical aliened graphene nanosheet acts as a reaction active spot. This 3D graphene-nanosheet-based doping approach can be applied to a wide variety of inorganic catalytic materials to enhance their performance and improve their durability in aspects such as single-atom catalysis and integration of multiple catalytic properties.

## 1. Introduction

Graphene, as an excellent two-dimensional carbon material carrier, has drawn enormous attention since Andre Geim and Konstantin Novoselov launched their micromechanical stripping method for large-scale graphene fabrication in 2004 [1,2,3,4,5,6,7,8]. In recent years, as one of the most exciting materials in current research, graphene has been widely used in various fields. Graphene’s high light transmittance and good electrical conductivity make it useful for light-emitting diodes [9,10,11], touch screens [12,13], and conductive electrodes [14,15,16]. Graphene and its derivatives are used for agricultural production and pollution management, such as crop mold detection, sewage purification, and gas detection [17,18,19,20]. As a nanocarrier, it is used for anticancer/gene delivery and cell culture in nanomedicine and biomedicine [21,22,23]. In the military and marine industrial fields, graphene-based materials are used as corrosion-resistant, wear-resistant, and flame-retardant protective coatings [24,25,26]. Especially, because of its excellent chemical catalytic properties, graphene is used extensively in the manufacture of various biochemical sensors for small biological molecule detection [27,28,29,30,31,32], which is vital for the early diagnosis of biologically related diseases [22,33,34,35,36]. Liu et al. fabricated a disposable cerium-oxide/graphene-nanosheet-based sensor to monitor acebutolol in environmental samples and biofluids [37]. The electrodes that they fabricated had a higher efficiency and conductivity. Xuan et al. developed a flexible glucose-graphene biosensor by depositing redox graphene on the surface of Ag/AgCl electrodes using microelectromechanical system fabrication technology and integrating a specific glucose oxidase onto the graphene surface [38]. Jing et al. fabricated a glucose graphene sensor based on zinc-oxide nanorods by hydrothermal method on a flexible polyester substrate [39]. The high conductivity of the electrodeposited reduced graphene promoted the transport of redox electrons and improved electron transport between the solution and the electrode significantly. Most of these graphene-based biochemical sensors use enzymes, antibodies, antigens, and other biomolecules with a corresponding substrate recognition ability as recognition element to convert chemical signals of the substance to electrical signals. Graphene-based biochemical sensors that rely on a variety of specific biological enzymes have been studied and applied extensively [40,41,42,43,44].

However, these oxidoreductases that grow on the graphene surface are affected by ambient temperature and light, and their properties are unstable and easily inactivated. This problem has led to a bottleneck in the improvement of the detection performance of graphene biochemical sensors and has hindered further improvements in detection sensitivity and detection limit [45,46,47,48,49]. To address these issues, we try to electrodeposit gold nanoparticles directly in situ on nanographene sheets to enhance the detection sensitivity and detection limit of the biochemical sensors.

Hereby, we fabricated a gold-boosted graphene AuNPs/GNSs/CC (gold nanoparticles/graphene nanosheets/carbon cloth) sensor to detect uric acid (UA) (Figure 1). The graphene grew directly on and bound closely with CC and did not fall off easily. The AuNPs were distributed uniformly on the AuNPs/GNSs/CC sensors. The GNSs/CC electrode with a three-dimensional (3D) structure possesses a large specific surface area (almost 2 cm^−2^ g^−1^), which provided sufficient hierarchical chemical active sites for AuNPs/GNSs/CC sensors. For a graphene growth time on the CC of 7 h, the GNSs/CC flexible electrode showed a good comprehensive detection performance for ascorbic acid (AA), dopamine (DA), and UA, respectively. The sensitivity of the AuNPs/7 h GNSs/CC electrode to UA was 6230 μA mM^−1^ cm^−2^, which is almost 6 times higher than that of the 7 h GNSs/CC electrode without gold deposition. The sensor was successfully used for UA detection in real human blood, and the recovery range is between 94% and 98%. This universal approach could be applied to various inorganic nano-catalytic materials to mimic inorganic nano-enzymes as the doping catalyst varied.

## 2. Material and Methods

### 2.1. Fabrication of AuNPs/GNSs/CC Sensor

To fabricate the AuNPs/GNSs/CC sensor (Figure 1), the thermal chemical vapor deposition (CVD) method was first used to fabricate GNSs/CC. CC (Cetech Co., Ltd., Taichung, China, WOS 1009) was placed in a tubular furnace, and argon (Ar), as a protective gas, was injected at 200 sccm and heated to 1100 °C in a tubular furnace to carbonize the CC fibers in an Ar atmosphere. The Ar flow was stopped after the carbon fiber was carbonized, and a mixed gas of H_2_ and CH_4_ was introduced at 160 and 6 sccm for 7 h. H_2_ was introduced to assist CH_4_ cracking and to generate in situ 3D graphene on a CC substrate through the thermal decomposition reaction of CH_4_, to complete vertical graphene growth. The GNSs/CC electrode material was obtained by re-introducing Ar to stabilize the graphene, drive away mixed gases of H_2_ and CH_4_, and cool the tube furnace to room temperature. To improve the catalytic capability of the electrode material, the AuNPs/GNSs/CC electrode was fabricated by electrochemical deposition of AuNPs on a 7 h GNSs/CC electrode (quenched by liquid nitrogen into 1 cm × 2 cm slices) with a comprehensive detection performance (see Section 3.2). The specific electrochemical deposition conditions were as follows: 0.6 mM HAuCl_4_ was used as the electrolyte, the 7 h GNSs/CC was used as the electroplating carrier, the scanning voltage was −0.2 V to 0.8 V, the scanning speed was 0.5 V s^−1^, and the scanning number was 6 to obtain AuNPs. Pending the end of electrochemical deposition, several washes with ultrapure water were used to wash away excess solution on the sensor for later detection.

By adjusting the carbon source introduction time from 0–10 h, CC electrodes with different graphene growing durations were fabricated to explore the effect of graphene thickness on the electrochemical property of the electrode sensor. These electrodes were termed 1 h GNSs/CC, 5 h GNSs/CC, 7 h GNSs/CC, and 10 h GNSs/CC electrodes, respectively.

### 2.2. Preparation of Experimental Related Solutions

Phosphate-buffered solution (PBS, 0.1 M) at pH 7.4 was used to investigate the electrochemical performance of the different electrodes. The PBS preparation method is described by using 500 mL as an example. Sodium dihydrogen phosphate (1.2 g, all required reagents are from Aladdin) and disodium hydrogen phosphate (5.68 g) were added to 500 mL of ultrapure water to yield PBS. The electrolyte solution-potassium ferricyanide/potassium ferricyanide solution (that contained 0.1 M KCl solution) for the measurement of electrochemical impedance was prepared by mixing a 0.01 M solution of K_3_[Fe(CN)_6_] and a 0.01 M solution of K_4_[Fe(CN)_6_]·3H_2_O. The HAuCl_4_ concentration for the deposition of AuNPs on the 7 h GNSs/CC electrode material was 0.6 mM, which was made from 30 mL of 0.5 M sulfuric acid solution and 180 μL of 0.1 M HAuCl_4_ solution.

### 2.3. Material Characterization

Scanning electron microscopy (SEM) and energy dispersive spectroscopy (EDS) characterization: The morphology, thickness, size, and distribution of AuNPs of the fabricated electrode materials were observed by using a scanning electron microscope (Hitachi, Tokyo, Japan, S-4700) at 15 kV. The cross section of the carbon fabric fiber was obtained by liquid nitrogen quenching. The material to be tested was removed after cooling in liquid nitrogen for 30 min and bent and clamped with tweezers to obtain a smooth cross section. The backscattered electron mode was used for elemental analysis of the energy spectrum.

X-ray photoelectron spectroscopy (XPS) characterization: The elemental compositions and combined states on the surfaces of two electrode materials, GNSs/CC and AuNPs/GNSs/CC, were detected by using a XPS instrument (ULVAC-PHI, Chigasaki, Japan, PHI 5000 VersaProbe II) with Al–Kα as the X-ray source.

Raman characterization: The characteristic peaks for different graphene strengths were characterized by micro-confocal Raman spectroscopy (HORIBA, Ltd., Kyoto, Japan, Confocal Raman microscopy). The testing parameters were within a detectable range of 1000–3000 cm^−1^, an integration time of 3 min, an integration number of 20, and a laser wavelength of 625 nm.

Brunauer–Emmett–Teller (BET) characterization: To explore the relationship between graphene thickness and specific surface area and the influence of graphene thickness on the electrochemical performance, we performed BET detection on the GNSs/CC electrodes by BET specific surface area analyzer (Quantachrome, Boynton Beach, Florida, USA, Autosorb-IQ-MP).

### 2.4. Electrochemical Testing

Electrochemical impedance spectroscopy (EIS) detection: Potassium ferricyanide/potassium ferrocyanide solution (0.01 mM) was used to test the electrochemical impedance spectroscopy of different electrode sensors at 0.1 Hz to 100 kHz, with an amplitude of 0.05 V and a standing time of 2 s.

Cyclic voltammetry (CV) detection: The influence of scanning rate on peak current was analyzed by cyclic voltammetry. The response of modified electrode with different thicknesses to three types of small molecules was studied and optimized by graphene CVD time. The experiments were carried out through electrochemical workstation (CHI 760 E). The working electrode was GNSs/CC electrode, and the counter electrode was platinum plate while the reference electrode was saturated calomel electrode. PBS (0.1 M) at pH 7.4 was used as the electrolyte solution with a scanning voltage varied between −0.6 V and 0.4 V.

Differential pulse voltammetry (DPV) detection: DPV was used to study the electrochemical property of the AuNPs/GNSs/CC electrode material. The linear range and sensitivity of the AuNPs/GNSs/CC electrode for detecting UA was obtained. The effect of gold nanoparticles on the catalytic capability of the electrode material was also investigated.

### 2.5. Detection in Real Blood Samples

The feasibility and accuracy of the AuNPs/GNSs/CC electrode sensor in detecting UA in human real blood samples was evaluated by comparing the test results with those obtained by dry chemical analysis in hospital. Human blood samples were collected from Peking University Shenzhen Hospital and tested and analyzed within 1 h after sampling. The collection of fresh human blood samples was approved by the hospital’s ethics committee and each volunteer provided informed consent.

## 3. Results and Discussion

### 3.1. Characterization of CC, GNSs/CC, and AuNPs/GNSs/CC

The surface of the CC fiber without chemical modification was bare and smooth, as shown in Figure 2a. After growing graphene on CC by chemical vapor deposition for one hour, sheet graphene appeared on the CC fiber surface (Figure 2b). Figure 2c shows that the graphene layer thickness increased with growth time. The complete graphene grew in the outer layer of the CC fiber when the growth time reached 5 h, with a thickness of 122 nm. When graphene was grown for 7 h, the graphene thickness was 134 nm. At a growth time of ten hours, graphene on the different fibers touched and filled the space between the fibers and formed a unique porous 3D graphene structure with a thickness of 322 nm. This special 3D reticular structure increased its specific surface area and contact area with small molecules and provided more active sites [50,51,52]. Gold nanoparticles were deposited on the graphene layer by electrochemical deposition to prepare the AuNPs/GNSs/CC sensor, which was characterized by SEM and EDS (Figure 2d). The presence of elemental C and Au on the AuNPs/GNSs/CC sensor and their uniform distribution indicated that AuNPs were deposited and distributed evenly on the GNSs/CC substrate, with the AuNPs size concentrated between 83.96 and 96.55 nm. Graphene with a 3D structure provides a reliable site for AuNPs attachment, which suggests that GNSs/CC is a suitable substrate for gold nanoparticles.

Raman spectroscopy characterization was performed on bare CC and GNSs/CC with different graphene growing durations (Figure 3a). Raman spectra of CC and GNSs/CC showed two characteristic peaks: a D peak at 1346 cm^−1^ (which represents defective, disordered or sp^3^ hybrid carbon atoms in graphene) and a G peak at 1588 cm^−1^ (which represents sp^2^ hybrid carbon atoms). The intensity of peak G explains the order degree of carbon atoms in graphene to a certain extent, whereas peak D represents the disorder degree of carbon atoms in graphene. A smaller I_D_/I_G_ peak strength ratio yields a smaller material defect and disorder [53]. In this work, the ID/IG ratio was 0.8 for CC and 0.7 for 10 h GNSs/CC, which indicates that the prepared graphene reduced the composite defects. Figure 3a shows that a longer graphene growth time on CC yields more obvious 2D peaks for GNSs/CC, which can be used to assess the successful fabrication of the graphene.

Table 1 shows the specific surface area of the GNSs/CC with different growing durations of graphene. Although the specific surface area of exposed CC was not obtained, the specific surface area increased significantly after graphene was grown on the CC. The specific surface area of GNSs/CC increased with graphene growth time (1 h: 1.3 cm^−2^ g^−1^, 5 h: 1.7 cm^−2^ g^−1^, 7 h: 1.9 cm^−2^ g^−1^, 10 h: 2.0 cm^−2^ g^−1^) because the graphene became thicker, denser, and formed a 3D structure, which increased the specific surface area of the carbon composite.

The element and chemical bond information of GNSs/CC and AuNPs/GNSs/CC electrode materials was obtained from XPS, and the test results are shown in Figure 3b. Compared with the XPS spectrum of GNSs/CC, the spectrum of AuNPs/GNSs/CC displays C 1s, O 1s, and Au 4f peaks, which indicates that gold nanoparticles were deposited on graphene substrates. Figure 3c shows the XPS spectra of the C 1S of two electrode materials. The peaks at 284.6, 286.5, 287.7, and 289.2 eV correspond to C-C, C-O, C=O, and O=C-OH bonds, respectively. Figure 3d is the XPS spectrum of AuNPs/GNSs/CC material for Au 4f. Peaks were visible at 84.1 and 88.1 eV, corresponding to the 4f_7/2_ and 4F_5/2_ peaks of Au, respectively, and this indicates that gold exists in the form of a simple substance.

The electrode conductivity is evaluated by contrasting the peak current with different modified electrodes in 0.01 mM [Fe(CN)_6_]^3−/4−^ and 0.1 M KCl solutions. Figure 3e displays the CV cures of CC, 7 h GNSs/CC, and AuNPs/GNSs/CC electrodes. Compared with CC, the response current of the 7 h GNSs/CC electrode to [Fe(CN)_6_]^3−/4−^ increased significantly, and the electron transfer capability of the CC electrode modified with graphene was better because the highly conductive graphene with 3D structure prepared by chemical vapor deposition increased the specific surface area of the electrode significantly and provided more active reaction sites. Compared with the GNSs/CC electrode, the AuNPs/GNSs/CC electrode exhibits a greater oxidation current in [Fe(CN)6]^3^^−/4^^−^ solution. The redox potential of gold nanoparticles is +1.68 V, which has a strong ability to capture electrons [54,55,56,57,58]. The addition of gold nanoparticles improves the conductivity and catalytic capability of the composites, and the large specific surface area of graphene has a good synergistic effect with gold nanoparticles.

The electrochemical impedance of different electrode sensors was tested (Figure 3f). The electrochemical impedance of the exposed CC electrode was approximately 245 Ω (curve a). When graphene was modified on the carbon fabric for 7 h, the impedance was reduced to 47 Ω (curve b). This result indicates that the GNSs/CC electrode had a faster electron transmission capacity, and therefore, a better electrochemical activity. When the Au nanoparticles were further modified on the CC electrode with graphene, the impedance value was reduced to 2 Ω (curve c), which indicates that the deposited gold nanoparticles and graphene have a good electrical conductivity, and the synergistic effect of the two can improve the electrical conductivity of the target electrode.

### 3.2. Detection Performance of 7 h GNSs/CC Electrode

We studied the catalytic capability of the GNSs/CC electrodes with different growth durations of graphene on CC for three small biological molecules by CV. Figure 4a shows the cyclic voltammogram of a 7 h GNSs/CC electrode in 0.01–4.0 mM AA and 0.1 M PBS solution. The oxidation potential of AA on the 7 h GNSs/CC is –68 mV. At different modified electrodes, the changes of oxidation peak potential and peak type of AA are not significant, and the peak current increases linearly with increasing the AA concentration. Figure 4d shows the fitting curve of the relationship between current and concentration obtained by detecting AA with GNSs/CC electrodes with different growth times of graphene. The electrochemical response and sensitivity of GNSs/CC to AA increased with increasing the graphene thickness. The maximum detection sensitivity was 244.3 μA mM^−1^ cm^−2^ at the 7 h GNSs/CC electrode, and the minimum detection concentration for AA was 0.001 mM. When the growth duration of the graphene was 5 h, a 3D graphene structure formed initially on the CC fiber (Figure 2c). The catalytic capability of graphene on AA improved when it was first generated. After the formation of the 3D reticular structure, its ability to gather reactive electrons was further improved and the active sites increased further. When the growth time of graphene on carbon fiber cloth was 7 h, the adsorption and enrichment of small molecules and the catalytic capability of 7 h GNSs/CC was maximized, and the detection sensitivity to AA was maximized. However, when the graphene growth time raised to 10 h, the detection sensitivity and response current of 10 h GNSs/CC to AA decreased, possibly because the graphene that was covered by the outer layer of CC fiber was too thick, and the graphene in the inner layer did not participate in the reaction. We used the GNSs/CC for detection of DA and UA and obtained the detection performance of GNSs/CC in 0.001–1.0 mM DA and 0.01–1.0 mM UA, respectively (Figure 4b,c). Figure 4e,f are the fitting curves of current and concentration that were achieved by detecting DA and UA with GNSs/CC electrodes with different growth times of graphene. An increase in graphene thickness resulted in a gradual increase in electrochemical response of GNSs/CC to DA and UA. The maximum detection sensitivity of the 7 h GNSs/CC electrode for detecting DA was 380 μA mM^−1^ cm^−2^, and the minimum detection concentration was 0.0001 mM. The maximum sensitivity of the 5 h GNSs/CC electrode for UA detection was 1872 μA mM^−1^ cm^−2^, and the minimum detection concentration was 0.001 mM.

The above analysis shows that the 3D graphene structure generated on the CC can increase the active sites on its surface; improve the adsorption enrichment and catalytic capacity of AA, DA, and UA; and thus improve the detection performance of the electrode for biological small molecules. However, compared with the other GNSs/CC with graphene growth at different times, 7 h GNSs/CC had the best comprehensive detection performance for the three biomolecules.

### 3.3. Effect of Deposition Sweep Number of Gold Nanoparticles on the Performance of AuNPs/GNSs/CC for 7 h GNSs/CC

The GNSs/CC electrodes could make a distinction between the redox peaks of the three biomolecules by CV and detect their existing contents quantitatively. However, a large non-Faraday current often occurs during the CV test, which leads to a large background current and limits the peak current drift [59]. Because of current difference reduction, the DPV method can reduce the background current that is caused by the redox current of the impurities, so it has a higher detection sensitivity and a lower detection limit. Therefore, we used the DPV method to detect single biological small molecules.

AuNPs have a high electron density, dielectric properties, and catalytic effect, and can be combined with various biological macromolecules without affecting their biological activity. The 3D graphene provides active sites for gold nanoparticles, and synergies with gold nanoparticles to improve the catalytic activity of the electrode materials. Therefore, an AuNPs/7 h GNSs/CC sensor with a good comprehensive detection performance was fabricated by electrochemical deposition of gold at the 7 h GNSs/CC electrode, and UA was detected (Figure 5). The detection performance of the AuNPs/7 h GNSs/CC sensor for UA is related to the number of deposition cycles of gold nanoparticles that were deposited on the 7 h GNSs/CC surface. Figure 5a–f shows the differential pulse curve and relationship curve between the current and concentration of UA that was detected by the AuNPs/7 h GNSs/CC with 2, 6, and 10 turns of gold nanoparticles deposition. The linear equations are *I*_P_(μA) = 3206 *C*_UA_ + 37.4, *I*_P_(μA) = 12,460 *C*_UA_ + 442.1, and *I*_P_(μA) = 51,700 *C*_UA_ + 169.5. The detection sensitivity was 1603, 6230, and 25,850 μA mM^−1^ cm^−2^, and the detection concentration range was 0.002–0.8, 0.005–0.2, and 0.002–0.04 mM, respectively. According to the above results, when the number of scanning cycles of gold deposition was 2, 2 segs AuNPs/7 h GNSs/CC had a wide detection range, but its detection sensitivity was poor. The 10 segs AuNPs/7 h GNSs/CC had a high sensitivity when the number of gold deposition scanning turns was 10, but its detectable concentration range was small. When the number of gold deposition scanning turns was 6, the 6 segs AuNPs/7 h GNSs/CC electrode had a better sensitivity and wider detection range. Subsequently, we used the 6 segs AuNPs/7 h GNSs/CC electrode for human blood sample detection. The response current and sensitivity of UA to AuNPs/7 h GNSs/CC electrode improved compared with that of ungilded GNSs/CC electrode. The sensitivity of 6 segs AuNPs/7 h GNSs/CC was nearly 6 times higher than that of GNSs/CC (Appendix A). Gold nanoparticles in the SEM analysis (Figure 2d) indicate uniform deposition in the graphene layer, because the small gold nanoparticles have a large specific surface area, which improves the active site of composite materials, the catalytic activity increases, and sedimentary modest gold particles can improve the sensitivity.

### 3.4. UA Detection of Human Blood Samples

Accurately detecting UA in human blood reflects the metabolic and immune functions in the human body, and therefore, it is important to test the performance of the 6 segs AuNPs/GNSs/CC sensor. We added serum samples with a known UA concentration to the buffer and recorded the peak current of the DPV response. The current value was inserted into the current and concentration fitting curve equation: *I*_P_(μA) = 12,460 *C*_UA_ + 442.1, the UA concentration in human blood samples was calculated, and UA concentration in raw serum was calculated according to serum dropwise addition. Table 2 compares the three groups of human serum samples with the AuNPs/GNSs/CC sensor and the results measured in the hospital. Compared with the hospital results, the recovery rate was between 94% and 98%, which indicates that the AuNPs/GNSs/CC sensor based on CC had a good recovery and accuracy in UA detecting of real human blood samples.

Table 3 lists six graphene-based sensors and their detection performance. It can be seen that graphene and its derivatives are very suitable as substrate materials for sensors due to their catalytic properties and excellent electrochemical activity. Compared with the graphene-based sensor without gold nanoparticles, our sensor has a better detection range and detection limit, because the addition of gold nanoparticles improves the conductivity and catalytic capacity of the composite. In addition, compared with other graphene-based sensors doped with gold nanoparticles, our sensor has a good linear detection range (UA: 5–200 μM) and a low detection limit (UA: 0.28 μM; S/N = 3).

## 4. Summary and Conclusions

In summary, we fabricated AuNPs/GNSs/CC sensors detecting UA by in situ depositing inorganic gold nanoparticles on nanographene sheets by electrochemical deposition. Compared with biological enzyme loading methods, our method solved the biological enzyme instability and improved the detection performance of graphene-based sensors. The 7 h GNSs/CC electrode had a large specific surface area of 1.9 cm^−2^ g^−1^, which provided sufficient chemically active sites for the AuNPs/GNSs/CC sensor. When the number of deposition turns of the gold nanoparticles on the GNSs/CC electrode is 6, the sensor had a good detection sensitivity and linear detection range. The sensitivity of the AuNPs/7 h GNSs/CC electrode to UA was 6230 µA mM^−1^ cm^−2^, which is almost 6 times higher than that of the 7 h GNSs/CC electrode without electroplating. The linear detection concentration range of the AuNPs/GNSs/CC sensor was 0.005–0.2 mM. The sensor was used in detecting UA in real human blood samples with recoveries from 94% to 98%, which shows the ability of the sensor to detect small biomolecules on a commercial basis. This approach is expected to be applied to a wide variety of inorganic nanocatalytic materials and graphene for composite loading and performance improvement, such as single-atom catalysis and integration of multiple catalytic properties.

## Figures and Tables

**Figure 1 nanomaterials-12-01327-f001:**
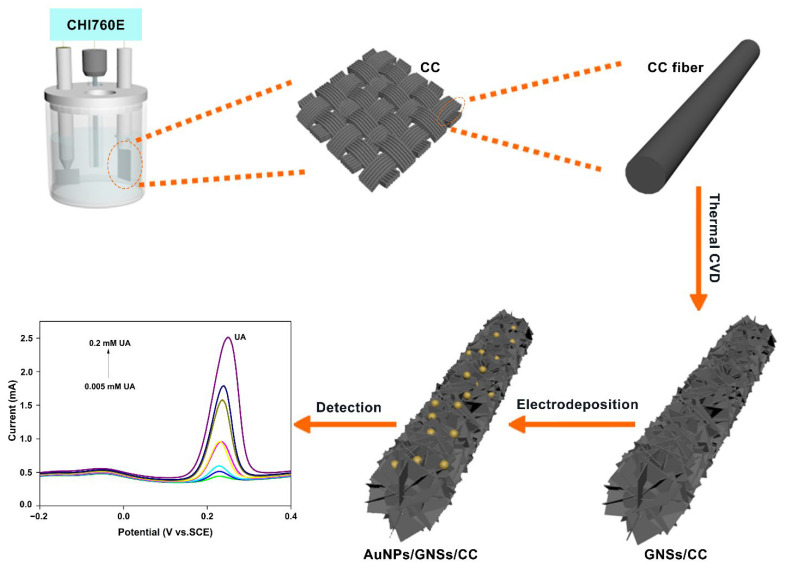
Fabrication of AuNPs/GNSs/CC sensor and application of UA detection. The UA concentration is varied between 0.005, 0.008, 0.01, 0.02, 0.04, 0.08, 0.1, and 0.2 mM.

**Figure 2 nanomaterials-12-01327-f002:**
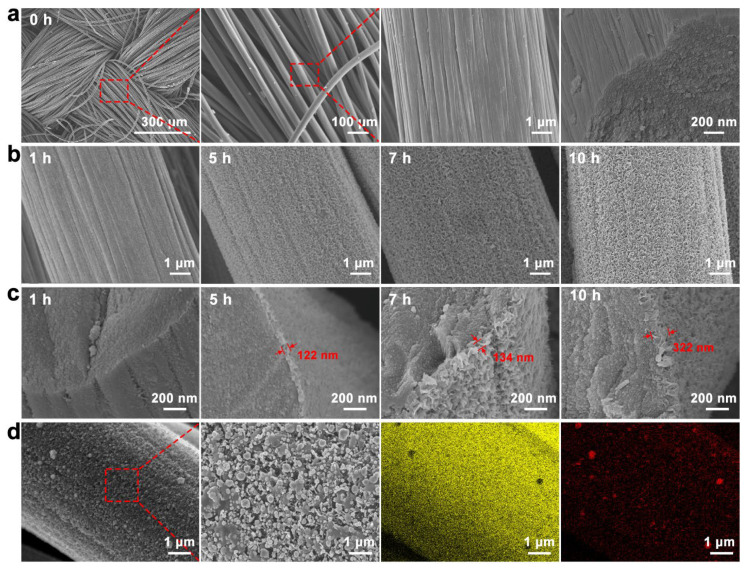
Basic characterization of CC, GNSs/CC, and AuNPs/GNSs/CC electrode materials. (**a**) SEM of exposed CC fiber. (**b**) SEM of GNSs/CC of graphene grown on CC at different times. (**c**) SEM of GNSs/CC cross sections of graphene grown at different times on CC. (**d**) SEM of AuNPs/GNSs/CC electrode and EDS of elemental Au and C on its surface.

**Figure 3 nanomaterials-12-01327-f003:**
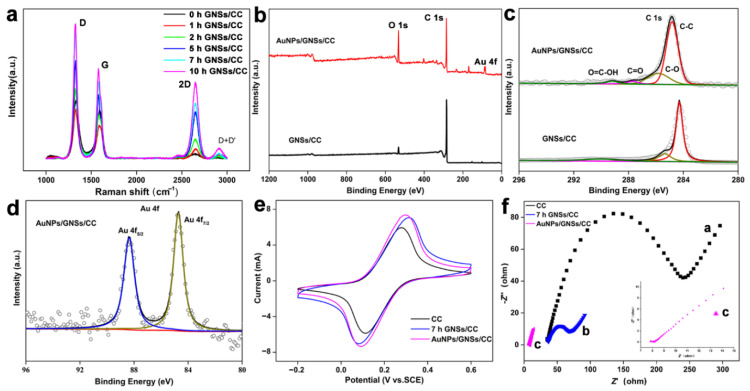
Basic characterization of different electrode materials. (**a**) Raman spectroscopy of bare CC and GNSs/CC with different growing durations of graphene. (**b**) XPS total spectrum of GNSs/CC and AuNPs/GNSs/CC. (**c**) C 1s spectra of GNSs/CC and AuNPs/GNSs/CC. (**d**) Au 4f spectra of GNSs/CC and AuNPs/GNSs/CC. (**e**) CV curves of CC, GNSs/CC, and AuNPs/GNSs/CC in 0.1 mM potassium ferricyanide/potassium ferrocyanide solution. (**f**) Electrochemical impedance spectroscopy diagram of CC, GNSs/CC, and AuNPs/GNSs/CC in 0.1 mM potassium ferricyanide/potassium ferrocyanide solution.

**Figure 4 nanomaterials-12-01327-f004:**
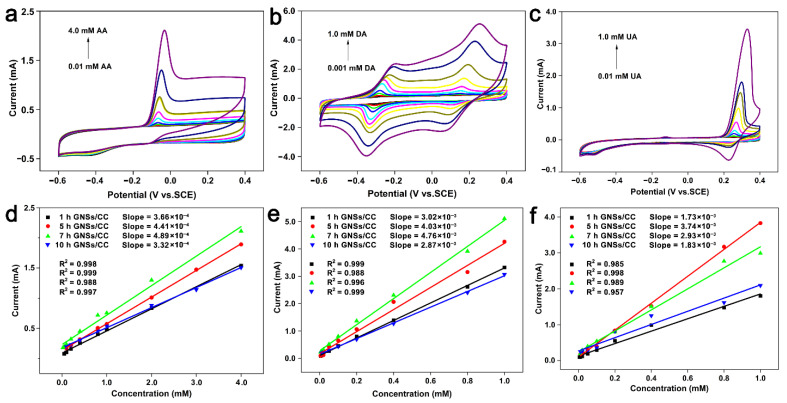
GNSs/CC electrodes are applied to detect (**a**) AA, (**b**) DA, and (**c**) UA by CV. Fitting of relationship between current and concentration of (**d**) AA, (**e**) DA, and (**f**) UA detected by different GNSs/CC electrodes.

**Figure 5 nanomaterials-12-01327-f005:**
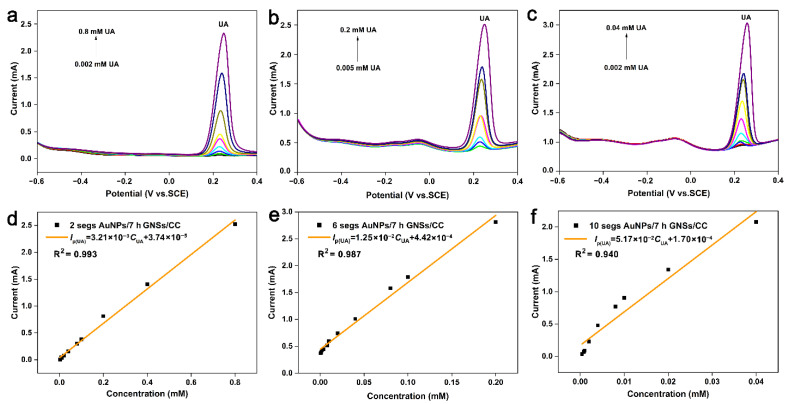
UA is detected by DPV method using AuNPs/7 h GNSs/CC deposited with different amounts of gold nanoparticles. DPV curve of UA detected by (**a**) 2 segs, (**b**) 6 segs, (**c**) 10 segs AuNPs/7 h GNSs/CC electrode. Fitting of current and concentration of UA detected by (**d**) 2 segs, (**e**) 6 segs, (**f**) 10 segs AuNPs/7 h GNSs/CC electrode.

**Table 1 nanomaterials-12-01327-t001:** BET of CC and graphene with different growth durations.

Sample	CC	1 h GNSs/CC	5 h GNSs/CC	7 h GNSs/CC	10 h GNSs/CC
BET (cm^−2^ g^−1^)	N/A	1.3	1.7	1.9	2.0

**Table 2 nanomaterials-12-01327-t002:** Detection of UA in real human blood samples.

Blood Sample	Results from the Hospital Tests (mM)	Results from Our Sensors (mM)	Recovery (%)
1	0.500	0.490	98
2	0.492	0.483	98
3	0.456	0.430	94

**Table 3 nanomaterials-12-01327-t003:** Performance comparison of sensors for UA detection.

Sensor	Linear Range (μM)	Detection Limit (μM)	Reference
MoS_2_-rGO	4–40	3.8	[60]
rGO-CNT/ITO	0.2–16	0.17	[61]
GNSs/CC	0.5–20	0.03	[59]
AuNPs/GO	10–800	206	[62]
rGO-PAMAM-MWCNT-AuNP	1–114	0.33	[63]
OPEDOT-AuNPs-ErGO/GCE	20–100	5	[64]
AuNPs/GNSs/CC	5–200	0.28	This work

## Data Availability

Not applicable.

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
