# Peer review of "Facile Gold-Nanoparticle Boosted Graphene Sensor Fabrication Enhanced Biochemical Signal Detection"

_nanomaterials, 2022, doi:10.3390/nano12081327_

Round 1

Reviewer 1 Report

In this study, gold-boosted graphene sensor was fabricated by the in-situ electrochemical deposition of gold nanoparticles on vertical graphene nanosheets. The novelty of this work is solving of the instability of biological enzymes and improving the detection performance of graphene-based sensors. The work is interesting and can be considered after addressing my issues:

-Almost all figures have low quality, please check and increase

-the reference style must be updated as Nanomaterial’s style

-I believe your paper can be enriched if you incorporate some new and interesting papers in the field of nanosensors: https://doi.org/10.1002/aelm.202100233, https://doi.org/10.1021/acs.analchem.9b03933, https://doi.org/10.1002/admi.202001978, https://doi.org/10.1007/s12274-021-3967-x and https://doi.org/10.3390/nano11061384

-Gold or Au NPs. Please be constituent in the whole manuscript

-some abbreviations were used without explanation

-please discuss your results with similar papers in this field

Reviewer 2 Report

Q1. P.1 Line.65: Why is it started the detail of experiment here? it should re-organize the manuscript. Moreover, the introduction is not enough contained of references for previous works and classification of previous works.

Q2. In figure 1, the explain of graph was not clear. Even I could not find each condition of line. Author just mentioned 0.005-0.2 mM UA but we do not know the detail of information. Figure should be modified with detail of information.

Q3. In figure 2-5, the resolution of figure is too low so it cannot see the letter in figure. 

Q4. In figure 4(d-f) and 5(d-f), what is the R2 value when plotting a line graph? Some lines look above 0.98.

Q5. Both figure 4 and 5 have same conditions, but the color of the lines is different. Match the same color as this can confuse the author.

Q6. Many studies on graphene-gold have been reported with a concept similar to this work. Why not compare the results? If the sensitivity or repeatability is better than these, it should be mentioned.

Round 2

Reviewer 1 Report

The paper fully improved in terms of discussion and results. by the way, I suggest the following articles be discussed as I mentioned before (https://doi.org/10.1021/acs.analchem.9b03933 and https://doi.org/10.3390/nano12060982) as these sensors gained lots of interests in recent years.

Reviewer 2 Report

I recommended that you accept this form, as the author has well modified the revised manuscript in the comments.

-Figure with high visibility

-Table for comparison

-Enough background in introduction

Thank you.
